# Comparative Pathobiology of the Intestinal Protozoan Parasites *Giardia lamblia*, *Entamoeba histolytica*, and *Cryptosporidium parvum*

**DOI:** 10.3390/pathogens8030116

**Published:** 2019-07-29

**Authors:** Andrew Hemphill, Norbert Müller, Joachim Müller

**Affiliations:** Institute of Parasitology, University of Berne, Länggass-Strasse 122, 3012 Berne, Switzerland

**Keywords:** diagnosis, immunopathology, intestinal infections, pathogens, treatment

## Abstract

Protozoan parasites can infect the human intestinal tract causing serious diseases. In the following article, we focused on the three most prominent intestinal protozoan pathogens, namely, *Giardia lamblia*, *Entamoeba histolytica*, and *Cryptosporidium parvum*. Both *C. parvum* and *G. lamblia* colonize the duodenum, jejunum, and ileum and are the most common causative agents of persistent diarrhea (i.e., cryptosporidiosis and giardiasis). *Entamoeba histolytica* colonizes the colon and, unlike the two former pathogens, may invade the colon wall and disseminate to other organs, mainly the liver, thereby causing life-threatening amebiasis. Here, we present condensed information concerning the pathobiology of these three diseases.

## 1. Introduction

Besides bacterial and viral pathogens, protozoan parasites can also infect the human intestinal tract and cause serious diseases [1]. In this article, we will focus on the three most relevant protozoal pathogens, namely, *Cryptosporidium parvum* (and closely related Coccidia), *Giardia lamblia*, and *Entamoeba histolytica*. We regard these organisms as obligate pathogens because they may cause symptoms in otherwise completely healthy individuals and disappear after clearance by the immune system and/or successful chemotherapy. Conversely, opportunistic pathogens are found in healthy individuals as a part of the normal microbiome and cause symptoms in challenged individuals only [2]. A (controversial) example is *Blastocystis hominis*, one of the most frequent eukaryotes isolated from feces [3] and pathogenic in immunocompromised and irritable bowel syndrome (IBS) patients [4]. Microsporidia belong to the kingdom of fungi [5] and would be the topic of a more detailed review on systemic mycoses.

*Giardia lamblia* and *C. parvum* are the most common pathogenic intestinal protozoan parasites and causative agents of persistent diarrhea in humans [6]. In the United States and in Europe, the annual incidences for the two parasitoses are around 10^4^ each. *Entamoeba histolytica* causes life-threatening amebiasis after invasion of the colon wall and advancement to the liver and other organs. In the EU and the US, most cases of amebiasis are associated with travelers coming from endemic areas. Worldwide, the annual incidence is estimated at around 100 million individuals (Table 1).

The three intestinal protozoans discussed in this review have very simple biological cycles. There are no intermediate hosts. Cysts or oocysts (*Cryptosporidium*) are excreted in feces and (auto) infection occurs via ingestion of these permanent stages (Figure 1).

As a consequence, due to the poor hygienic conditions, the prevalence of these three intestinal protozoans is, however, much higher in underdeveloped countries thereby constituting a major problem for global health (see Table 2 for an overview).

## 2. Etiology and Epidemiology

### 2.1. Giardia lamblia

*Giardia lamblia* is an anaerobic, but to some extent also aerotolerant, eukaryote with several prokaryotic properties [7,8,9] belonging to the phylum Diplomonadida, super-group Excavata [5]. *Giardia* exists in two morphologic forms: the multi-flagellated trophozoite (four pairs of flagella) and the cyst. The trophozoite is dinucleated, pear-shaped, multi-flagellated, 9 to 15 µm long, 5 to 15 µm wide, and 2 to 4 µm thick, with an adhesive disk on the ventral surface (Figure 2). 

Trophozoites live attached on dudodenal and jejunal epithelial cells and thrive on nutrients from the intestinal fluid with amino acids, especially arginine as their preferred fuel [10,11]. Detached trophozoites form quadrinucleated, thick-walled cysts (8–10 µm in diameter). The cysts are excreted in the feces and constitute the infectious stage. Thus, giardiasis is caused by fecal contaminations of drinking water [12], food [13], or direct contact with feces [14], waterborne transmission being regarded as a major source [15]. Pathogenesis and virulence depend on the genotype of the *Giardia* strain and the immune and nutritional status of the host. To date, eight genetic assemblages (A to H) have been described. It is well established that isolates from assemblages A and B cause infection in humans [16,17,18]. There is, however, more recent evidence that isolates from assemblage E are pathogenic for humans, as well [19]. Since these strains can be found in humans as well as in animals, giardiasis can be regarded as a zoonosis [20]. Especially, young children living in poor sanitary conditions are exposed to giardiasis which—in combination with malnutrition or immunosuppression (e.g., HIV)—can be fatal. Moreover, persistent infections in children may cause stunting [21].

### 2.2. Entamoeba histolytica

Human amebiasis is caused by *E. histolytica* (Amoebozoa, Amorphaea). All pathogenic *Entamoeba* are classified as *E. histolytica*, whereas the species *Entamoeba dispar* comprises non-pathogenic *Entamoeba* strains [22]. As for *G. lamblia*, two stages can be distinguished, namely, trophozoites and cysts. The motile, mononucleated trophozoite (10 to 20 µm ᴓ, sometimes larger) colonizes the colon (and eventually other organs), where it may transform into the cyst stage having a similar size, but one to four nuclei (Figure 3).

Human infection occurs via the ingestion of excreted cysts, thus via fecal–oral contamination upon waterborne, foodborne, or person-to-person transmission. Therefore, poor sanitation and overcrowding are socio-economic factors favoring amebiasis. In the EU and the US, nowadays, cases of amebiasis are mostly associated to travelers coming from endemic areas. Amebiasis is still a major cause of morbidity and mortality in developing countries [23].

### 2.3. Cryptosporidium sp.

The genus *Cryptosporidium* is classified into the phylum Apicomplexa, class Conoidasida, and order Eucoccidiorida. More recent studies, however, indicate, that this genus is more closely related to Gregarines [24]. Currently, 31 valid *Cryptosporidium* species have been recognized in fish, amphibians, reptiles, birds, and mammals, and an additional 40 distinct genotypes from a variety of vertebrate hosts are described. Nearly 20 species and genotypes have been reported in humans, of which *C. hominis* and *C. parvum* account for >90% of all cases [25]. *Cryptosporidium hominis* transmission occurs via humans, while *C. parvum* has a high zoonotic potential. There is still a lack of subtyping tools for many *Cryptosporidium* species of public and veterinary health importance, and the genetic determinants of host specificity of *Cryptosporidium* species are only poorly understood. Diarrhea is caused through mechanisms involving increased intestinal permeability, chloride secretion, and malabsorption. Although otherwise healthy individuals can acquire infection, other conditions such as immunodeficiency caused by HIV infection, malnutrition, chemotherapy, diabetes mellitus or bone marrow or solid organ transplantation constitute increased risks for more severe and disseminated disease [26]. Livestock, particularly cattle, are important reservoirs of zoonotic infections [27]. The impact is especially devastating in infants in the resource-constrained regions, and disease is associated with an estimated annual death rate of >200,000 children below 2 years of age. An epidemiological study of over 22,000 infants and children in Africa and Asia [28] found that *Cryptosporidium* was one of the four pathogens responsible for most of the severe diarrhea and was considered the second greatest cause of diarrhea and death in children after *Rotavirus* [29]. Common means of transmission of cryptosporidiosis is by municipal drinking water and water in swimming pools and via contaminated food. A high risk of infection concerns child care workers, parents of infected children, people who handle infected animals, those exposed to human feces through sexual contact, and healthcare providers as well as pregnant women and individuals suffering from immunodeficiency [26].

*Cryptosporidium* has a monoxenous life cycle [30]. Infection takes place via oral ingestion of oocysts containing invasive sporozoites. These sporozoites enter intestinal epithelial cells and form a parasitophorous vacuole (PV) that is located at the apical part of the host cell, just underneath the brush border (Figure 4 and Figure 5).

These sporozoites will then develop into merozoites, which either undergo asexual proliferation, egress, and re-infection of other intestinal cells (named type I merozoites), or develop into type II merozoites and undergo sexual development, leading to the formation of a zygote, which produces an oocyst wall and infective sporozoites. In the case that the oocysts are thin-walled, excystation of sporozoites can take place already in the intestine, leading to auto-infection, while thick-walled oocysts are excreted at very large numbers and are infectious upon oral ingestion. Oocysts are environmentally stable and can survive for many months under temperate and moist conditions, and they are resistant to chlorine levels usually applied in water treatment.

## 3. Pathogenicity and Virulence

### 3.1. General Remarks

The virulence of an intestinal pathogen results from its own genetic background, the competence of the host immune system, its nutritional status (e.g., the acquisition of iron [31]) and the interaction with other intestinal microorganisms (Figure 6).

All three parasites discussed here in detail infect their hosts via ingested cysts and colonize the digestive tract. They attach to the epithelial surface of the duodenum/ileum (*Cryptosporidium* sp., *Giardia lamblia*) or colon (*Entamoeba histolytica*) and elicit an immune response involving interleukin (IL)-6 production by T-cells, dendritic cells and mast cells. Interleukin-6 stimulates IL-17-mediated host defense (production of intestinal IgA (Immunoglobulin A) and anti-microbial peptides). Furthermore, mast cell degranulation promotes peristalsis. The resulting inflammatory reactions (see Reference [32] for review) are more (*E. histolytica*) or less (*G. lamblia*) pronounced. Thus, immunopathology may play an important role in the case of *E. histolytica*, where inflammatory reactions may facilitate the penetration of the colon wall and the subsequent systemic spreading causing amebiasis. In the case of giardiasis and cryptosporidiosis, inflammatory reactions are much less pronounced.

### 3.2. Giardia lamblia

Unlike other protozoal pathogens, *G. lamblia* trophozoites and their conversion into cysts can be easily studied in vitro, the genome is sequenced (GiardiaDB.org), molecular genetics are well established, and in vivo-models (e.g., orally cyst-infected mice) are available. Nevertheless, to date, the pathophysiology of giardiasis is not entirely understood [33]. When ingested cysts reach the stomach, their cyst wall is digested and they transform into trophozoites colonizing the epithelium of the duodenum and the proximal jejunum. As a first reaction on part of the host, cells in contact with the trophozoites may undergo apoptosis, epithelial tight junctions are ruptured [34], and CD8+-lymphocytes are activated [35]. As a consequence, brush-border microvilli are shortened [35], resulting in deficiencies in disaccharidases and other enzymes [36]. In a next step, adaptive immune responses are elicited via IL-6-producing dendritic and mast cells [37] and CD4+ T-cells producing IL-17 and TNFalpha [38]. Another potential source of IL-17 are tuft cells that may be stimulated by metabolites excreted by trophozoites thereby enhancing the pro-inflammatory reactions in the intestinal epithelium, as found for closely related protists and for helminths [39]. As a consequence, cytotoxic [40] secretory IgA and defensins are produced resulting in the elimination of trophozoites from the intestinal surface. Moreover, nitric oxide (NO) produced by epithelial cells and immune cells inhibits trophozoite proliferation [33,41] and—NO production in neurons—in combination with mast cell degranulation—promotes intestinal peristalsis thus contributing to the expulsion of trophozoites. Although minor, the intestinal microbiota may increase the efficiency of the anti-giardial immune response. In fact, trophozoites may induce dysbiosis of the microbiota resulting in an immunological effect supporting infection [42] (Figure 7). Recent studies consider cysteine proteases involved in pathogenesis, disruption of intestinal epithelial junctions, apoptosis of intestinal epithelial cells, and also the degradation of host immune factors (e.g., immunoglobulins and chemokines), mucus depletion, and microbic dysbiosis as major virulence factors [18].

Since IgA recognizes surface proteins as predominant antigens, *G. lamblia* has developed an escape strategy based on the variation of these surface proteins. According to a generally admitted hypothesis, only one (major) variant surface protein (VSP) is expressed on a single trophozoite [43]. The expression of different VSPs—and thus antigenic variation—is triggered by epigenetic mechanisms involving changes in the chromatin state [44] and/or RNA interference [45,46]. Trophozoites surviving the exposure to IgA react by expressing different variants of these so-called “variant surface proteins” thereby escaping the immune response [47]. This strategy is called “antigenic switch” (Figure 8).

Antigenic switch does not occur only as a reaction to exposure to antibodies, but also to drugs [48,49,50] and is—in all likelihood—epigenetically regulated [51]. Recent results suggest that VSPs play not only a passive role in escaping host immune responses but also may actively participate in damaging epithelial cells via proteolytic activities [52]. An excellent review on parasitic strategies to circumvent host immune reactions is given elsewhere [32].

### 3.3. Entamoeba histolytica

*Entamoeba histolytica* trophozoites are easily cultivated in vitro, the genome has been sequenced (AmoebaDB.org), and susceptible and resistant rodent in vivo models are available, thus allowing experimental investigations of virulence factors [53,54]. While passing the digestive tract, ingested cysts liberate trophozoites proliferating within the colon. Unlike *Giardia*, they dwell not only on intestinal fluids but produce cysteine proteases and Gal/GalNAc-lectins both damaging the intestinal mucosa by structural destabilization and cellular destruction of the epithelial cell layer. As a consequence, *E. histolytica* penetrates the intestinal mucosa by evading and, at the same time, exploiting the mucosal immune response of the host. In an initial phase, mucosal inflammation is promoted by secretion of *E. histolytica* macrophage migration inhibitory factor (EhMIF). Supported by tissue-destructive and cytolytic effectors such as matrix metalloproteinases (MMPs) and oxygen free radicals (ROS) produced by infiltrating inflammatory cells, focal perforation of the intestinal mucosa may occur allowing the trophozoites to invade the colon wall [55,56]. 

Infecting *E. histolytica* trophozoites, however, have to face different host defense mechanisms, namely, (i) increased mucus production protecting the epithelial surface; (ii) secretion of defensin 2 and pro-inflammatory cytokines (Il-1β, IL-8, TNF-α) after contact of trophozoites with epithelial cells [57]; (iii) Th1-mediated immune responses during acute amebiasis, and Th2- and Th17-mediated immune responses during chronic amebiasis [58]. Consequently, neutrophils are attracted via IL-8 and macrophages via IL-1β. IL-17 production favors persistence of infection as shown by comparing IL-17-knock-out to wild-type mice [59]. Since *E. histolytica* trophozoites resist killing by neutrophils, the resulting inflammatory reaction even enhances tissue injury thereby promoting the infection [60]. In 90% of patients, the colon wall is not invaded and the infection remains asymptomatic or with mild symptoms. In the remaining 10%, the colon barrier is broken, and trophozoites spread into the wall and surrounding tissues causing local necrosis and ulcer formation. 

Once the colon wall is invaded, however, amebiasis may spread hematogenously to any organ in the body, most commonly the liver and the lungs [60,61]. It is still unclear to which extent host cells are directly involved in the destruction of the colon wall. There is some evidence from in vivo models that the inflammatory reactions of host cells and not proteolytic degradation of the wall by the parasite is responsible for tissue damage, but it is difficult to extrapolate from defined animal models to the situation in patients with diverse physiological backgrounds and immune status [62].

Another intriguing aspect is the interaction with the colon microbiome. Enteric bacteria may stimulate the oxidative stress defenses of *E. histolytica* [63]. Moreover, there are good indications that *E. histolytica* causes dysbiosis of the colon microbiome stimulating immune responses facilitating systemic invasion [64]. It is still unclear to which extent this parasite evades or suppresses [65] host immune responses during this systemic spread (Figure 9).

### 3.4. Cryptosporidium sp.

Unlike *G. lamblia* and *E. histolytica*, *Cryptosporidium* sp. (and other closely related Coccidia, e.g., *Eimeria* sp.) cannot be cultivated axenically [66]. The genome has been sequenced (CryptoDB.org), but genetic manipulation of this protozoal parasite [67] has only recently been established using CRiSPR/Cas-mediated gene targeting [68]. The studies leading to the identification of pathogenicity and virulence factors are performed in coculture systems [69,70] with intestinal [71], biliary [72], tracheal [73] or esophagus [74] epithelial cell layers or organoids [75], where infection, invasion, and differentiation can be studied from a couple of days until several weeks [76]. Based on studies with these systems, factors mediating excystation, invasion, PV formation, intracellular maintenance, and host cell damage could be identified as important virulence factors [77]. After oral uptake of oocysts, *Cryptosporidium* sporozoites hatch in the small intestine, attach to and invade host epithelial cells [78], preferentially cells in mitosis [79]. This is in good agreement with more recent findings that in pig intestinal cells infected with *C. parvum*, expression of genes involved in mitosis are upregulated, but neither stress- nor apoptosis-related genes [80]. Host-cell apoptosis is dimmed in early stages of infection and promoted in late stages [81,82,83]. In analogy to related apicomplexan parasites, it can be speculated that *Cryptosporidium* secretes molecules to the host cell that directly interfere with apoptosis signaling cascades at multiple levels as shown for *Toxoplasma gondii* [84] and for *Theileria* sp. [85,86].

Attachment to host cells and invasion is mediated by proteins such as a galactose/N-glactosamin-specific lectin [87] and *T. gondii*-SAG1 homologous proteins [88] released from a set of secretory organelles, which are constituents of the apical complex. Infection requires extensive remodeling of the cytoskeleton [89], and invaded sporozoites are finally located in a parasitophorous vacuole, surrounded by a parasitophorous vacuole membrane (PVM) that is essentially host cell surface membrane derived and modified after invasion. Once intracellular, parasites remain at the apical domain of the host cell, therefore, the PV is an intracellular but extracytoplasmatic compartment (Figure 5). Thus, the PVM shields parasites from the host defense. Trophozoites are formed and will eventually undergo development into merozoites, as well as sexual development forming sporozoites and oocysts, which are orally infective. Tissue damage occurs through disruption of tight junctions of the epithelium, leading to cytoskeletal alterations, loss of barrier function, and—ultimately—increased cell death. These alterations are caused by lytic enzymes such as phospholipases [90] and proteases [91,92,93,94]. 

As a response, proinflammatory cytokines like interferon gamma (IFNγ) [95], IL-6, IL-12 [96], IL-17 [97], and chemokines are released to the infection site. A detailed review is given in Reference [98] and the references therein. On the one hand, the resulting inflammatory reactions contribute to increased epithelial permeability, impaired intestinal absorption, and increased secretion thereby enhancing the symptoms like malabsorption and diarrhea. On the other hand, they stimulate innate defense mechanisms [95] such as the release of antimicrobial peptides [99] and acquired immune responses leading ultimately to the control of the infection in immunocompetent hosts (see Figure 7). 

## 4. Control and Treatment

### 4.1. Prevention

Since all three pathogens discussed in this review are transmitted via cysts [100], the prevention is water treatment by filtration [101] or disinfection by chlorination or radiation [102]. Moreover, person-to-person and animal-to-person transmission can be prevented by standard attention to hygiene (i.e., hand washing). Treatment of asymptomatic persons excreting cysts is indicated to prevent autoinfection and the spread of infection to healthy persons.

### 4.2. Diagnosis

The standard methodology as well as recent developments in diagnosis of intestinal parasites including *G. lamblia*, *E. histolytica*, and *C. parvum* have recently been summarized in the updated guidelines from the Infectious Diseases Community of Practice of the American Society of Transplantation [103]. Conventionally, diagnosis is based on the detection of trophozoites or (oo)cysts in stool samples. In addition, *E. histolytica* trophozoites can be identified in aspirates or biopsies sampled during colonoscopy or surgery, respectively. Coprological diagnosis of cryptosporidiosis is mostly performed by modified Ziehl–Neelsen staining of acid-fast *Cryptosporidium* oocysts in fecal smears (see Figure 4). This method is regarded as the gold standard method for diagnosis of this protozoan parasite. While serology is considered to be of minor importance in diagnosis of giardiasis and cryptosporidiosis, ELISA tests demonstrating seroconversion as an indicator of an invasive *E. histolytica* infection are important tools to diagnose acute amebiasis.

Detection of ameba in stool is, however, not conclusive since non-pathogenic *Entamoeba* strains (*E. dispar*) cannot be distinguished from pathogenic strains by a mere morphological examination. This problem is solved by various *E. histolytica*-specific PCR tests [104,105]. A modern multiplex real-time PCR-based method allows the simultaneous detection of *Giardia*, *Entamoeba,* and *Cryptosporidium* in stool samples with high sensitivities and specificities [106]. These tests can be employed not only on stool, but also on biopsy material where trophozoites are rarely visible.

### 4.3. Treatment

Like for other anaerobic pathogens, the first line treatment of amebiasis and giardiasis—as recommended by the WHO and CDC guidelines—is based on compounds containing a nitro group which is reduced, thereby forming toxic intermediates, metronidazole [107] or related nitroimidazoles being the first choice (see Table 2). The nitro-thiazolide nitazoxanide is active not only against *Giardia* (in vitro and in vivo) but also against *Entamoeba* in vitro [108]. Since nitazoxanide and other thiazolides affect host cells also [109], they are not suited for the treatment of systemic infections such as amebiasis. In the case of resistance against nitro drugs which is frequent in *Giardia* [110], the highly effective benzimidazole albendazole may be used as a second line drug [111]. Many natural compounds, such as essential oils, some fatty acids, isoflavones, etc., inhibit the proliferation of *G. lamblia* trophozoites with much higher efficacy than metronidazole (see References [112] and references therein). Therefore, a diet combining low protein contents resulting in less free amino acids, especially arginine, available as fuel [11] with traditional herbs, such as basil, garlic, ginger, oregano etc., may constitute an additional management strategy, especially in the case of chronic giardiasis.

Resistance to nitro drugs also occurs in *E. histolytica* [113]. Therefore, there is a constant interest in novel, druggable targets that are absent in the host [114]. An example for such a target is the metabolism of sulfur-containing amino acids, e.g., methionine gamma-lyase-mediated catabolism [115]. Since virulence of *E. histolytica* strains is directly correlated to their oxygen resistance [116], the inhibition of scavengers of reactive oxygen species such as thioredoxin reductase [117] by auranofin, as identified by an unbiased high throughput screening [118] or by related compounds, could constitute a complementary strategy [119]. 

Against cryptosporidiosis, chemotherapy would be valuable in immunocompromised patients, but an effective regimen has not been established [26]. Nitazoxanide and other non-nitro-thiazolides are effective against cryptosporidiosis in a rodent model [120], but it is unclear whether the compounds act on the parasite or on the host [109]. For some HIV-infected patients, paromomycin, an oral non-absorbed aminoglycoside from *Streptomyces rimosus*, or paromomycin combined with azithromycin (a macrolide antibiotic) may be at least partially beneficial [121]. In some studies, treatment with nitazoxanide significantly shortened the duration of diarrhea and decreased the risk of mortality in malnourished children [122], and clinical trials showed efficacy in adults [123]. However, other studies showed that nitazoxanide therapy failed to exhibit activity in immunocompromised patients [124,125]. Since the studies showing effectiveness in patients come from one sole source, it is questionable whether there is at all efficacy of nitazoxanide against cryptosporidiosis in patients.

More promising, elevating CD4+ T-cell levels in HIV-infected patients by highly active antiretroviral therapy has led to the cessation of life-threatening cryptosporidial diarrhea [79]. This improvement is likely to result from immune reconstitution but may, in part, also be due to the antiparasitic activity of protease inhibitors. Thus, for therapy in HIV patients, nitazoxanide or paromomycin combined with azithromycin should be considered along with initiating antiretroviral therapy [79,83]. Currently, more promising compounds are being developed [126], including inhibitors of inosine monophosphate dehydrogenase [127], calcium-dependent protein kinase I [128], and lipid kinase [129] as the most striking examples.

### 4.4. Is Vaccination a Suitable Strategy?

Over the last two centuries, vaccination has proven most useful in fighting against a plethora of life-threatening infective agents. Therefore, it is reasonable that many groups invest in developing vaccination tools and strategies against intestinal protozoan pathogens.

In the case of giardiasis, these attempts are hampered by “antigenic switch” (see above). To circumvent this problem, a potential vaccine strain expressing multiple variant surface proteins (VSPs) on a single cell has been created by disrupting the RNA interference mechanism silencing VSP expression [130]. Allegedly, experimentally infected gerbils can be protected against subsequent infections by “vaccination” with this strain or by immunization with recombinant VSPs.

Cryptosporidiosis can be prevented by vaccination with lyophilized oocysts, as shown for calves [131]. This result is of particular interest for the poultry industry where the closely related *Eimeria* constitute a major problem [132]. As a consequence, there is a constant and continuing effort to identify suitable candidates such as, for example, protein p23 [133], a rhomboid protein [134], or surface glycoproteins [135] as vaccine candidates, either as recombinant proteins, DNA vaccine or expressed in a suitable bacterial strain [136].

Vaccination strategies against both diseases may work in suitable animal models and may even have some relevance for farm animals, but in the case of human patients, they are hampered by the fact that the subpopulations with the highest exposure and the highest risk such as malnourished children and HIV-positive adults are immunocompromised. As consequence, the immune responses elicited by recombinant vaccines are impaired [98]. A successful vaccine candidate against giardiasis should comprise the whole pattern of VSPs that may be expressed on an infective strain. This is impossible since the genomes of the sequenced *Giardia* strains contain several hundred VSP genes, and not all pathogenic strains are amenable to culture and, thus, to sequencing and characterization of their VSP patterns.

Clearly, vaccination seems to be a more promising strategy against amebiasis [137]. Vaccination with the Gal-lectin [138] successfully prevents amebiasis in gerbils [139,140], baboons [141], and as a subunit vaccine in a mouse model [142]. Moreover, other suitable vaccine candidates have been identified in the *Entamoeba* genome [143]. The challenge is to pass from a suitable animal model to tests in humans.

## 5. Conclusions and Outlook

Taken together, intestinal protozoan pathogens, in particular *G. lamblia*, *Cryptosporidium* sp., and *E. histolytica*, cause disabling and even life-threatening diseases. Transmitted by cysts, they can be controlled through water treatment. Antigiardial chemotherapy is well established and alternatives to the main lines of treatment are available. Against cryptosporidiosis, suitable treatment options are, however, lacking and constitute an interesting field for future investigations.

Amebiasis, the most lethal of the three, is also the most difficult to treat. Here, the development of both chemotherapy and vaccination will be one of the most challenging issues in parasitology for the future. Another interesting field for investigations would be the influence of overcome giardiasis, cryptosporidiosis, and infection by non-invading ameba on the gastrointestinal absorption of nutrients, the microflora, and the immune system [31].

## Figures and Tables

**Figure 1 pathogens-08-00116-f001:**
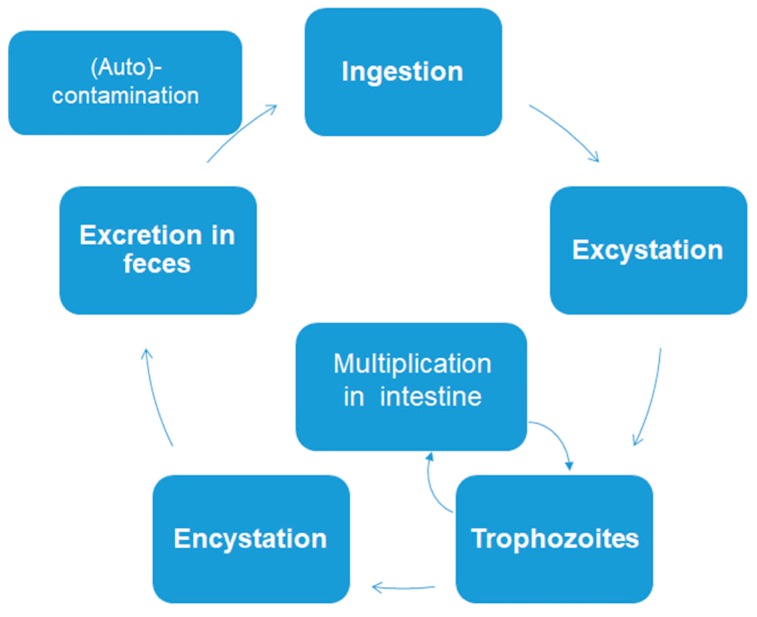
Simplified biological cycle of Giardia lamblia, Entamoeba histolytica, and Cryptosporidium parvum.

**Figure 2 pathogens-08-00116-f002:**
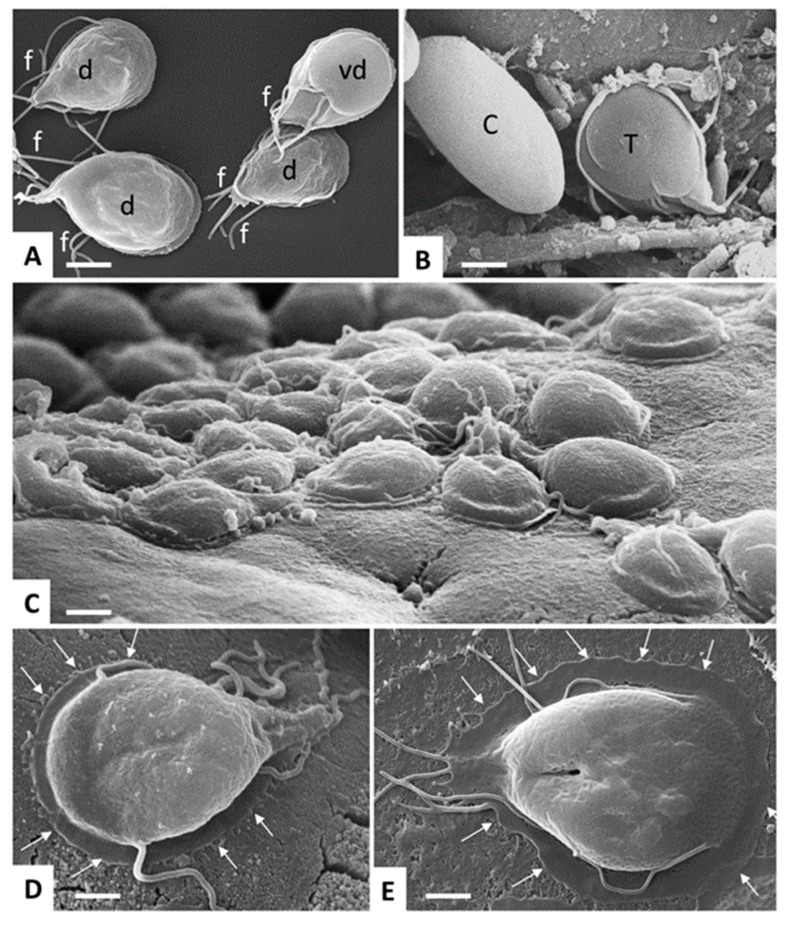
*Giardia lamblia* trophozoites and cysts visualized by SEM. (**A**) trophozoites cultured in axenic in vitro culture, exposing either their dorsal surface (d) or ventral disc (vd). Note the multiple flagella (f). Bar = 8 µm. (**B**) Trophozoite (T) and cyst (C) stages in a mouse feces sample. Bar = 6.4 µm. (**C**) Trophozoites attaching to the intestinal surface of an experimentally infected mouse. Bar = 8 µm. (**D**,**E**) Higher magnification view of a trophozoite attaching to the mouse intestinal surface and to human colon carcinoma cells (Caco2) during in vitro culture, respectively. Arrows delineate the periphery of the ventral disc. Bar in (**D**,**E**) = 4 µm.

**Figure 3 pathogens-08-00116-f003:**
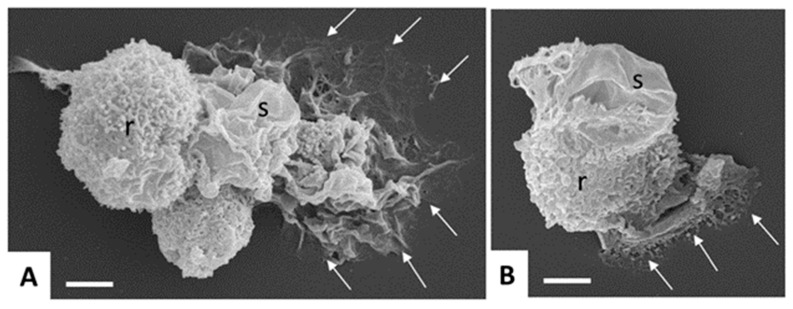
Scanning electron microscopy of in vitro-cultured *Entamoeba histolytica* trophozoites. Note the different shapes and cell surface structures such as rough (r) and smooth (s) adopted by the trophozoites, and the cytoplasmic protrusions mediating contact to the surface (arrows). Bar in (**A**,**B**) = 5 µm.

**Figure 4 pathogens-08-00116-f004:**
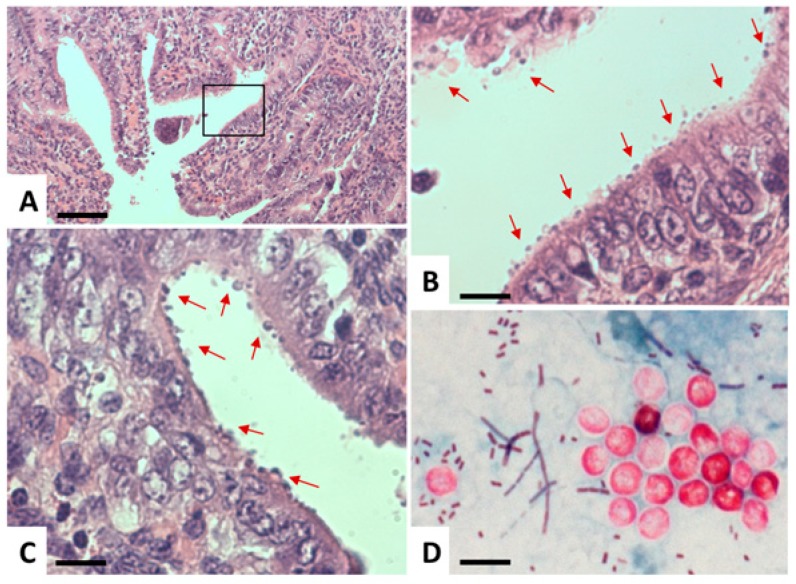
Light microscopy of *Cryptosporidium*. (**A**–**C**) Histological sections of paraffin-embedded *C. parvum-*infected intestinal tissue, stained with hematoxylin and eosin. (**A**) A low-magnification view, where the boxed area in (**A**) is magnified in (**B**). Arrows point towards *C. parvum* parasitophorous vacuoles seen as round bodies on the surface of the epithelial layer. (**D**) Modified Ziehl–Neelsen staining of oocysts (red, circular bodies) in a stool sample. Bar in A = 1450 µm; B and C = 260 µm; D = 7.5 µm.

**Figure 5 pathogens-08-00116-f005:**
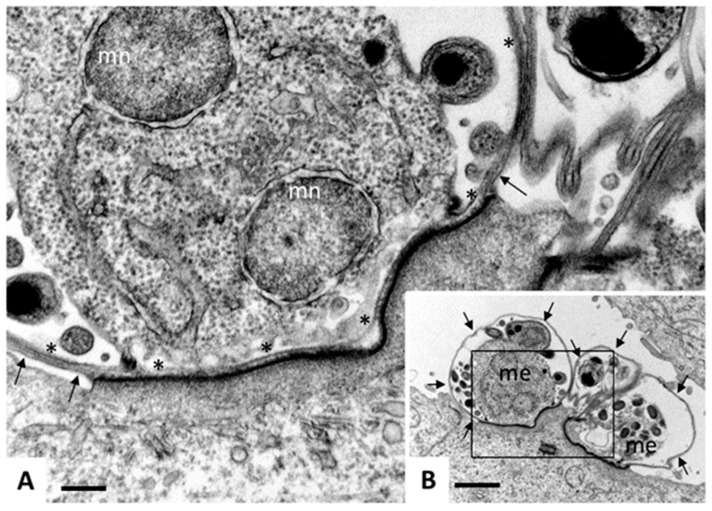
Transmission electron microscopy (TEM) of *Cryptosporidium parvum*. Madin Darbey canine kidney (MDCK) cells were infected with *C. parvum* sporozoites and fixed and processed for TEM after 72 h of culture. Sporozoites have formed parasitophorous vacuoles (PVs) on the apical part of the MDCK cells, occupying a space which is still intracellular, but essentially extra-cytoplasmatic, giving rise to meronts. (**B**) A low magnification view of three PVs, with two developing meronts (me) clearly visible. Arrows indicate the outer host cell surface membrane. Bar = 12 µm. (**C**) A higher magnification view of the boxed area in (**A**). Asterisks (*) indicate the membrane of the parasitophorous vacuole, arrows point towards the host cell surface membrane, mn indicates nuclei of developing merozoites. Note the electron-dense zone where the PV is in close contact to the host cell cytoplasm, formed due to the cytoskeletal rearrangements. Bar = 2.5 µm.

**Figure 6 pathogens-08-00116-f006:**
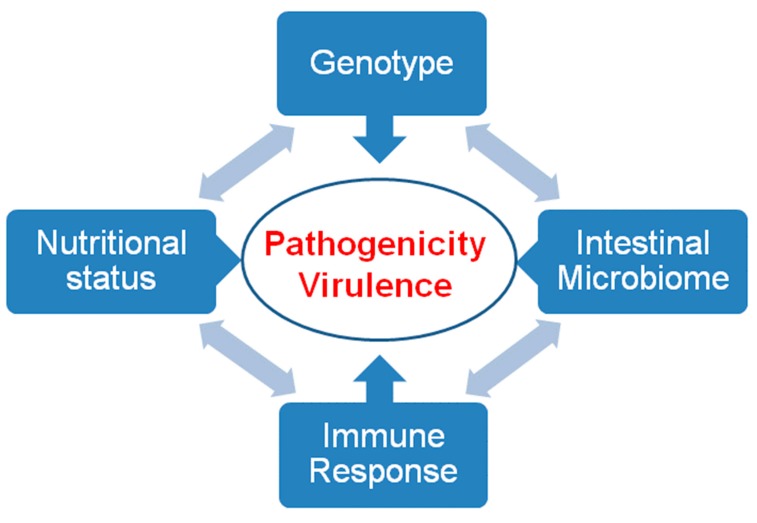
Pathogenicity and virulence. Pathogenicity and virulence are the result of interactions between the genotype of the pathogen, the immune response, the nutritional status, and the intestinal microbiome of the host.

**Figure 7 pathogens-08-00116-f007:**
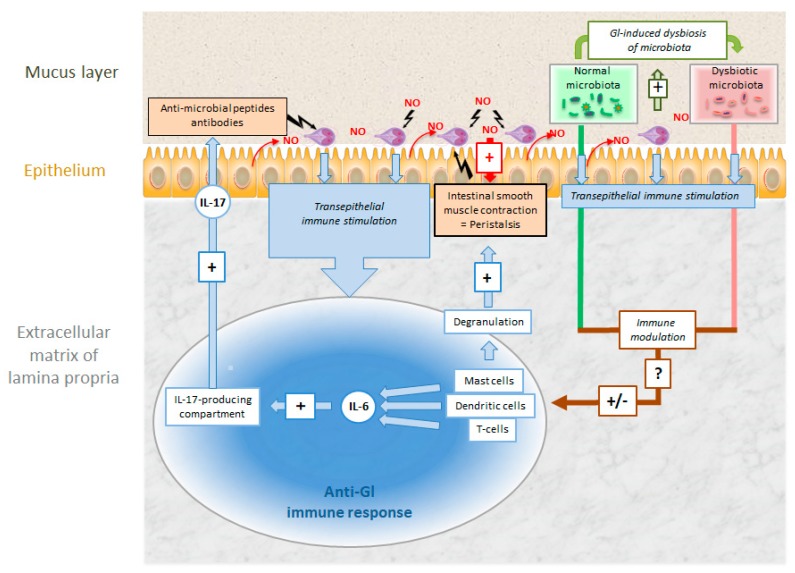
Intestinal colonization of *Giardia lamblia* (Gl) and host defense mechanisms against the parasite. Explanation see text.

**Figure 8 pathogens-08-00116-f008:**
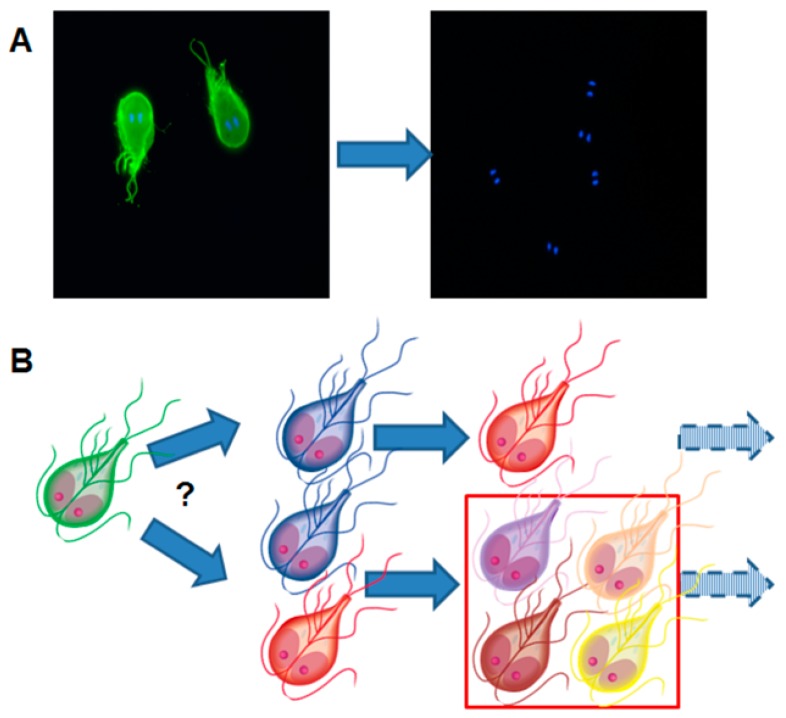
Illustration of the fundamental paradigm of antigenic variation in *Giardia lamblia*. (**A**) Trophozoites are stained with a monoclonal antibody directed against the variant surface protein (VSP) H7 (green staining). The presence of trophozoites is visualized by the characteristic staining of the double nuclei. After in vivo culture and re-isolation, the green staining is lost, meaning that new VSPs have replaced VSP H7. (**B**) Is one VSP replaced by another VSP or by several different VSPs thereby increasing the heterogeneity of the trophozoite population? Transcriptional studies have revealed that after subsequent in vivo cultivation of *G. lamblia* clone H7, the variability of VSPs increases.

**Figure 9 pathogens-08-00116-f009:**
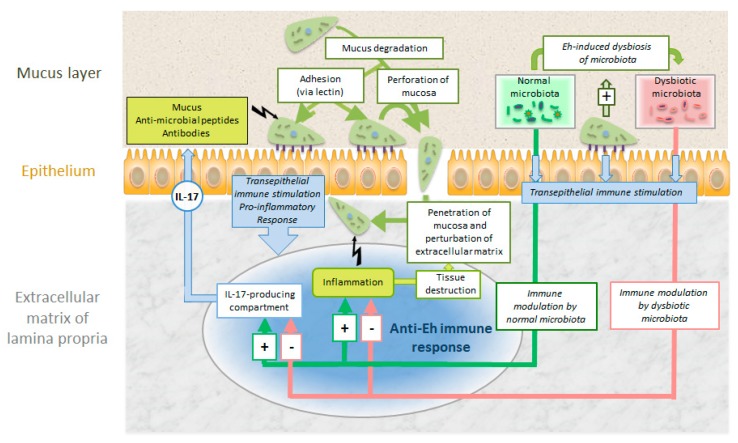
Intestinal colonization and invasion of *E. histolytica* (Eh) and host defense mechanisms against the parasite. For explanations see text.

**Table 1 pathogens-08-00116-t001:** Protozoa causing intestinal infections. The protozoa presented in this review are in bold.

**Species**	**Classification** **(Super Groups)**	**Incidence**			**Pathogenicity**	**Localization**
		**World ^1^**	**US ^2^**	**EU ^3^**		
*Balantidium coli*	Cliliata (Diaphoretickes)	Rare				colon
*Blastocystis* sp.	Stramenopile (Diaphoretickes)	Very high			opportunistic (?)	colon
***Cryptosporidium parvum***	**Apicomplexa** **(Diaphoretickes)**	**nk**	**8–9**	**7**	**obligate**	**duodenum, jejunum, ileum**
*Dientamoeba fragilis*	Trichomonadina (Excavata)	Common			unclear, most likely same as *G. lamblia*	colon
***Entamoeba histolytica***	**Amoebozoa** **(Amorphea)**	**100**	**rare**	**rare**	**obligate**	**colon, liver**
***Giardia lamblia***	**Diplomonadida** **(Excavata)**	**250**	**15**	**18**	**obligate**	**duodenum, jejunum, ileum**
*Microsporidia* sp.	Fungi (Amorphea)	Very high			opportunistic	colon

^1^ WHO (World health organization), NIH (National Institute of Health) (×10^6^/year); ^2^ CDC (Center of Disease Control), data for 2011–2012; ^3^ ECDC (European Center of Disease Control), data for 2014–2015 (both ×10^3^/year). nk, not known. Websites: ecdc.europa.eu; www.cdc.gov; www.nlm.nih.gov; www.who.int.

**Table 2 pathogens-08-00116-t002:** Overview of diseases caused by the protozoans presented in this review; see Reference [1] and text of this review for further references.

	Giardiasis	Amebiasis	Cryptosporidiosis
Pathogen	*Giardia lamblia*	*Entamoeba histolytica*	*Cryptosporidium parvum*
Transmission	Via (Oo)cysts in feces		
Symptoms			
Acute	Persistent diarrhea (>1 w), malabsorption.	Diarrhea, abdominal pain.	Mild-to-acute diarrhea, nausea, abdominal pain, low-grade fever.
Chronic	Malabsorption, loose stools, gassiness, cramping, fatigue, liver or pancreatic inflammations.	Fever, sepsis, liver abscesses, skin lesions.	Severe diarrhea, vomiting, malabsorption, volume depletion and wasting, biliary and respiratory involvement in immunodeficient persons.
Diagnosis			
Feces Biopsy material	Microscopy (cysts), coproantigen test, PCR.	Microscopy (trophozoites, cysts), coproantigen test and PCR.	Microscopy, coproantigen test, PCR, enzyme-immunoassays.
Serology		Positive in the case of extraintestinal infection.	
Differential Diagnosis	Cryptosporidiosis, IBS, celiac.	IBD, cancer, bacterial infections.	Giardiasis, *Rotavirus*, *Cyclospora cayetanensis*, *Clostridium difficile.* Microsporidia, IBS, celiac.
Management			
First line treatment	Metronidazole (500 to 750 mg p.o. t.i.d., 10 d)		Immunocompetent: NTZ (nitazoxanide)100–500 mg p.o. twice daily, 3 d. HIV: Antiretroviral therapy, possibly combined with NTZ or paromomycin/azithromycin. Other immunodeficiencies: NTZ 500 mg twice daily, 14 d.
Prevention	Personal hygiene, water treatment, appropriate cleaning and storage of vegetables.		

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
