# Peer review of "Comparative Pathobiology of the Intestinal Protozoan Parasites Giardia lamblia, Entamoeba histolytica, and Cryptosporidium parvum"

_pathogens, 2019, doi:10.3390/pathogens8030116_

Round 1
Reviewer 1 Report
This is a review of the immunopathology of 3 important intestinal parasites. It is not particularly well-written or well-referenced in many places.
Line 26. The argument for why these organisms are “obligate pathogens” is not made very well. While I agree with this conclusion, criteria 1 is rather circular and criteria 2 is vague and not supported with any references or data.
Table 1 should include jejunum as a site for giardia and cryptosporidium infection. The incidence column is confusing as the sub-columns are inconsistent. The footnotes need better references for these data.
Table 2 is not referred to in the text, only in the legend for table 1. This table reads from top-to-bottom instead of across which is the preferred style. Include IBS and Celiac as differential diagnoses for Giardia. Did authors mean inflammatory bowel disease (IBD) rather than irritable bowel syndrome (IBS) for differential diagnosis for amebiasis?
Table 3 is also only referred to within table 1.
Line 58. Giardia is clearly a eukaryote, and is no longer viewed as having “prokaryotic properties”. It has some unusual features for a eukaryote based on its long evolutionary divergence, but this description is not useful. Also, the phylum is Diplomonadida I believe.
Line 77. Reference 15 is a poor choice to support this statement. A few authors have reported assemblage E in humans now as well. A good review should discuss the impact of such new findings.
Line 81. Giardiasis in humans is rarely considered to be fatal. It is a major contributor to stunting however. See Rogawski and the MAL-ED group.
Lines 146-51 are redundant with the preceding paragraphs.
Line 158. Rather than reference another review, it would be much better to refernce the original GEMS study (Kotloff et al.) or the revised study based on quantitative PCR (Liu et al.).
Lines 173-80 lack any references and the description of the immunology is over-simplified and pertains better to Giardia than either to Cryptosporidium or Entamoeba. Immunity against Cryptosporidium requires IL-12 and IFNg much more than IL6 and IL-17.
Line 198. delete “dendritic and mast cells”.
Lines 198-200. Tuft cells haven’t been studied in the context of these pathogens. Delete or clarify so as not to mislead the reader.
Figures 6 and 7 are largely redundant.
Figure 8A. These two images do not appear to be the same field. As such, they do not actually demonstrate antigenic variation.
Line 241. A reference is needed for a role for IL-17 in amebiasis. Overall, the role of cytokines in amebiasis is not clear-cut. This review would benefit from a thorough analysis of the data rather than presenting a summary model that is not well-supported.
Similarly, the section on C. parvum ignores the role of IL-12 and IFNg. It also downplays the recent advances in C. parvum genetic manipulation and in vitro culture systems. Several of these have been published since this review was submitted, and the authors should update this section accordingly.
Author Response
Reviewer 1 (comments in green)
This is a review of the immunopathology of 3 important intestinal parasites. It is not particularly well-written or well-referenced in many places.
We hope that we could improve the quality in our revised version.
Line 26. The argument for why these organisms are “obligate pathogens” is not made very well. While I agree with this conclusion, criteria 1 is rather circular and criteria 2 is vague and not supported with any references or data.
We have rewritten the sentence and removed points i. and ii. although they are quite clearly understandable in our mind. Relevant references are 1 and 2.
Table 1 should include jejunum as a site for giardia and cryptosporidium infection. The incidence column is confusing as the sub-columns are inconsistent. The footnotes need better references for these data.
Table 1 has been amended accordingly (see also reviewers 2 and 3). Concerning the incidence numbers, we have included the most information given by public organisations that we could find. In the future, dramatic changes of these numbers are not expected.
Table 2 is not referred to in the text, only in the legend for table 1.
We do not agree in this point. Table 2 is referred to in the text between table 1 and 2 (not in a footnote of table 1) and in chapter 4.3.
This table reads from top-to-bottom instead of across which is the preferred style. Include IBS and Celiac as differential diagnoses for Giardia. Did authors mean inflammatory bowel disease (IBD) rather than irritable bowel syndrome (IBS) for differential diagnosis for amebiasis?
These corrections have been made.
Table 3 is also only referred to within table 1. – Table 3 has been deleted.
Line 58. Giardia is clearly a eukaryote, and is no longer viewed as having “prokaryotic properties”. It has some unusual features for a eukaryote based on its long evolutionary divergence, but this description is not useful. Also, the phylum is Diplomonadida I believe.
Giardia is clearly a eukaryote, no doubt, but – with all due respect - has many prokaryotic properties, f. i. its gene expression system which therefore can be targeted by antibacterial antibiotics, pathways of energy and intermediate metabolism etc. These properties are summarized in the reviews 7 and 8 as quoted. For more clarity, we have included an older, but still pertinent review by the Upcrofts stressing this point.
Line 77. Reference 15 is a poor choice to support this statement. A few authors have reported assemblage E in humans now as well. A good review should discuss the impact of such new findings.
There are also less poor choices of references to support our statement. They have been included now in the text (refs 17 and 18). Moreover, we have included one reference concerning assemblage E in humans (ref 19; ll. 80ff).
Line 81. Giardiasis in humans is rarely considered to be fatal. -not in otherwise healthy, but in malnourished and immunocompromised patients. See text.
It is a major contributor to stunting however. See Rogawski and the MAL-ED group. A corresponding reference (21) has been inserted (l. 85).
Lines 146-51 are redundant with the preceding paragraphs.
The chapter has been restructured accordingly (ll.115 ff).
Line 158. Rather than reference another review, it would be much better to reference the original GEMS study (Kotloff et al.) or the revised study based on quantitative PCR (Liu et al.).
Do you mean Sow et al., 2016? We have included this reference now.
Lines 173-80 lack any references and the description of the immunology is over-simplified and pertains better to Giardia than either to Cryptosporidium or Entamoeba. Immunity against Cryptosporidium requires IL-12 and IFNg much more than IL6 and IL-17.
The idea was to give a simplified overview. References are included in the subchapters.
Line 198. delete “dendritic and mast cells”.
Corrected.
Lines 198-200. Tuft cells haven’t been studied in the context of these pathogens. Delete or clarify so as not to mislead the reader.
Not in the context of Giardia, but in the context of other protists and helminths. We have clarified this point (ll.198ff).
Figures 6 and 7 are largely redundant.
Fig.6 has been removed.
Figure 8A. These two images do not appear to be the same field. As such, they do not actually demonstrate antigenic variation.
In fact, they cannot come from the same field since antigenic variation occurs in living trophozoites upon subcultures or as a response to treatments and not in fixed ones on the slide. In other words, it is absolutely impossible to demonstrate antigenic variation on the same trophozoites.
Line 241. A reference is needed for a role for IL-17 in amebiasis. Overall, the role of cytokines in amebiasis is not clear-cut. This review would benefit from a thorough analysis of the data rather than presenting a summary model that is not well-supported.
The role of cytokines in amebiasis is not well understood and clearly merits further investigation. A thorough analysis of the published data would need a review by its own. A reference for the ambiguous role of IL-17 in amebiasis has been inserted in the line, as suggested.
Similarly, the section on C. parvum ignores the role of IL-12 and IFNg. It also downplays the recent advances in C. parvum genetic manipulation and in vitro culture systems. Several of these have been published since this review was submitted, and the authors should update this section accordingly.
We do not intend to downplay any advances. So far, there is no axenical culture system available. Our statement is therefore correct. Moreover, we have quoted a quite complete list of coculture systems (already in the first version!) including a recent reference (74). Concerning genetic transformation, efforts have been made. with Crispr/Cas as the most outstanding advance (as quoted). To underline this aspect, an additional reference (67) has been added to the manuscript. To underline the roles of IFN-gamma and IL-12, additional references (95, 96) have been added. Furthermore, a detailed review for readers interested in these specific aspects had been quoted already in the first version of this manuscript.
Reviewer 2 Report
This manuscript entitled “Immunopathology of infections by the intestinal protozoan parasites Cryptosporidium parvum, Entamoeba histolytica and Giardia lamblia” is a needed summary of immunopathology of giardiasis, amoebiasis and cryptosporidiosis. This is a potential interesting review.
Some questions the authors might consider:
1. In the title the sequence of parasites is different from the text.
2. The authors could consider include biological cycles of parasites in a figure.
3. The process of auto-infection in cryptosporidiosis should be mentioned/discussed.
4. The lines 34 and 35 could be deleted.
5. The table 3 could be deleted since it list 4 websites and 3 genomes sites; it could be included in the text or references.
6. Please specify in line 60: multiflagellated trophozoite (4 pairs of flagella)
7. The authors repeated incidence and prevalence of the diseases many times in the manuscript (table 1, lines 79, 155, etc).
8. The figure 5 is confusing: parasite´s genotype? Can nutrional status alter genotype? (please explain in the figure legend).
9. In the figure 7 please correct: NO seems to be produced in the mucus layer.
10. It is important to discuss more details of the structural and functional abnormalities observed in experimental and human diseases. Please specify when immunopathological manifestations were observed in animal models or humans.
11. E. histolytica: how host cells are directly involved in destruction of colonic epithelial cells? Please discuss. For tissue damage ischemic due to inflammation see Pérez-Tamayo et al., Arch Med Res. 2006;37(2):203-9 and Kantor et al., Can J Gastroenterol Hepatol. 2018.
12.The authors could discuss the level of immunopathological damage in these parasitic diseases and the DTH reaction involved in these lesions.
13. See Sorci G et al., Pathogens. 2013; 2(1):71-91, for a discussion related to immunopathology and parasitic strategies to escape of immune system.
14. The discussion of data was good; I would have liked to see some views on a way forward. Which others issues in immunopathology of these parasitic diseases can be addressed in the future?
Author Response
Reviewer 2 (comments in red)
Comments and Suggestions for Authors
This manuscript entitled “Immunopathology of infections by the intestinal protozoan parasites Cryptosporidium parvum, Entamoeba histolytica and Giardia lamblia” is a needed summary of immunopathology of giardiasis, amoebiasis and cryptosporidiosis. This is a potential interesting review.
Some questions the authors might consider:
1. In the title the sequence of parasites is different from the text.
The title has been modified accordingly.
2. The authors could consider include biological cycles of parasites in a figure.
The cycles of the three parasites are similar and very simple. A scheme illustrating this point has been inserted as a new figure 1 (ll.47 ff). The other figures have been renumbered accordingly (2 to 9 now).
3. The process of auto-infection in cryptosporidiosis should be mentioned/discussed.
It is mentioned in the text (l. 318) and in the new figure 1.
4.The lines 34 and 35 could be deleted.
It is unclear which sentence is concerned. We have therefore left the text at this place as it was.
5. The table 3 could be deleted since it list 4 websites and 3 genomes sites; it could be included in the text or references.
Table 3 is deleted and the information is placed in the text.
6. Please specify in line 60: multiflagellated trophozoite (4 pairs of flagella).
The correction has been made.
7. The authors repeated incidence and prevalence of the diseases many times in the manuscript (table 1, lines 79, 155, etc).
This has been corrected, now. Please consider, however, that Tables 1 and 2 summarize information that is contained in the review. Therefore, there is and will be redundancy between the text and these two tables.
8. The figure 5 is confusing: parasite´s genotype? Can nutrional status alter genotype? (please explain in the figure legend).
Of course not. The figure schematizes interactions between genotype, nutritional status etc. and pathogenicity and virulence of a parasite. We have modified the figure, and hope that it is clearer, now. Moreover, we have chosen a different title for this chapter in order to minimize confusions (see also points 10 and 12).
9. In the figure 7 please correct: NO seems to be produced in the mucus layer.
The scheme is corrected.
10. It is important to discuss more details of the structural and functional abnormalities observed in experimental and human diseases. Please specify when immunopathological manifestations were observed in animal models or humans.
Immunopathological manifestations are observed in the case of amebiasis and not (or to a much lesser extent) in the case of the two other diseases discussed here. To avoid misunderstandings, we have modified the title and chapter 3.1. accordingly.
11. E. histolytica: how host cells are directly involved in destruction of colonic epithelial cells? Please discuss. For tissue damage ischemic due to inflammation see Pérez-Tamayo et al., Arch Med Res. 2006;37(2):203-9 and Kantor et al., Can J Gastroenterol Hepatol. 2018.
Chapter 3.3. has been amended accordingly (ll. 243-250 and 262-266).
12.The authors could discuss the level of immunopathological damage in these parasitic diseases and the DTH reaction involved in these lesions.
See points 8, 10, 11. Unlike in diseases caused by helminths, delayed-type hypersensitivity reactions do not play a significant role in these processes, at least not to our knowledge.
13. See Sorci G et al., Pathogens. 2013; 2(1):71-91, for a discussion related to immunopathology and parasitic strategies to escape of immune system.
This excellent review is introduced now l.179 and l.235.
14. The discussion of data was good; I would have liked to see some views on a way forward. Which others issues in immunopathology of these parasitic diseases can be addressed in the future?
As mentioned above, immunopathology is an important issue in amebiasis and to much lesser extents in giardiasis and cryptosporidiosis. An open issue would be the influence of .giardiasis, cryptosporidiosis and non-invading ameba on the gastrointestinal absorption of nutrients, the microflora, and the immune system. A corresponding remark has been added (ll. 419 ff).
Reviewer 3 Report
The MS submitted is an excellent review of the three main protozoa responsable of diarrhea commonly detected in travelers with a cosmopolitan distribution worldwide.
However this reviewer considers that some aspects could be revised to improve the quality of the revision:
1) This reviewer considers mandatory to revise throughout the manuscript the names of genus and species of the parasites and write them in italics (for example: species of table 1, lines 57, 59, 75, 77, 78, 100, 102, 105, etc). However "sp." should NOT be in italics (lines 99, 260)
2) The latin words like in vitro or in vivo should be also be written in italics (for example: line 64, 189, etc).
3) This reviewer suggests another type of separation in the incidence data of Table 1 or to include a line when the data of World, US, EU epidemiology is not known, to avoid possible confusion for the readers.
4) References in table 2 should be included
5) Table 3 is unnecessary.
6) I consider that the paragraph related with diagnosis needs to be more precise and make a more in-depth development, including the conventional staining methods, the immunochromatographic, fluorescent and enzyme immunoassays. I believe that this modifications will improve the quality and interest of the manuscript.
7) In the treatment epigraph, this reviewer suggests to include the recommendations made by the WHO and CDC in its treatment guidelines.
Author Response
Reviewer 3 (comments in blue)
The MS submitted is an excellent review of the three main protozoa responsible of diarrhea commonly detected in travelers with a cosmopolitan distribution worldwide.
However this reviewer considers that some aspects could be revised to improve the quality of the revision:
1) This reviewer considers mandatory to revise throughout the manuscript the names of genus and species of the parasites and write them in italics (for example: species of table 1, lines 57, 59, 75, 77, 78, 100, 102, 105, etc). However "sp." should NOT be in italics (lines 99, 260).
The corrections have been made., as well as the corrections of other misspellings.
2) The latin words like in vitro or in vivo should be also be written in italics (for example: line 64, 189, etc).
Idem.
3) This reviewer suggests another type of separation in the incidence data of Table 1 or to include a line when the data of World, US, EU epidemiology is not known, to avoid possible confusion for the readers.
Table 1 has been amended accordingly.
4) References in table 2 should be included
Table 2 gives a summary of the review. The references are included in the subchapters.
5) Table 3 is unnecessary.
Table 3 has been deleted.
6) I consider that the paragraph related with diagnosis needs to be more precise and make a more in-depth development, including the conventional staining methods, the immunochromatographic, fluorescent and enzyme immunoassays. I believe that this modifications will improve the quality and interest of the manuscript.
The chapter on diagnostics (4.2.) has been rewritten.
7) In the treatment epigraph, this reviewer suggests to include the recommendations made by the WHO and CDC in its treatment guidelines.
The recommendations made by these institutions are listed in table 2. This point has been emphasized in chapter 4.3 (l. 341).
Round 2
Reviewer 1 Report
This version is much improved, and I think remaining issues are due to incorrect English.
Line 195. "Another potential source of IL-17..." is a better way to phrase this since tuft cells have not been investigated in giardiasis.
Line 200. ... and NO production by neurons - in combination with mast cell degranulation - ...
Lines 249-54 should be clarified:
Line 250: What drives the mucous production by epithelial cells? IL-17 or infection in general?
Line 251-2. These responses 9according to the cited reference) are independent of IL-17 , but the sentence is confusing in this regard.
Line 252-3. Point 3 also doesn't elucidate IL-17 mediated responses as implied by lines 249-50.
Line 277 would be better:
and genetic manipulation of this protozoal parasite has recently been established using CRiSPR/cas mediated gene targeting.
Line 305-6 should be clarified. epithelial cells don't really make IL-12 or IFNg. IL-6 could be made by epithelial cells or myeloid cells. IL-12 likely comes from myeloid cells and IFNg and IL-17 probably come from T cells and ILCs, but not epithelial cells as is written.
Author Response
Reviewer 1
This version is much improved, and I think remaining issues are due to incorrect English.
Line 195. "Another potential source of IL-17..." is a better way to phrase this since tuft cells have not been investigated in giardiasis.
Corrected in the text as indicated.
Line 200. ... and NO production by neurons - in combination with mast cell degranulation - ...
Corrected in the text as indicated.
Lines 249-54 should be clarified:
Line 250: What drives the mucous production by epithelial cells? IL-17 or infection in general?
Line 251-2. These responses (according to the cited reference) are independent of IL-17 , but the sentence is confusing in this regard.
Line 252-3. Point 3 also doesn't elucidate IL-17 mediated responses as implied by lines 249-50.
We agree that the lines are unclear and have reformulated them. The former reference 57 underlining the role of IL-17 in the persistence of infection is now introduced in a separate sentence (l. 254-255; now as ref. 59)
Line 277 would be better:
and genetic manipulation of this protozoal parasite has recently been established using CRiSPR/cas mediated gene targeting.
We have reformulated this sentence.
Line 305-6 should be clarified. epithelial cells don't really make IL-12 or IFNg. IL-6 could be made by epithelial cells or myeloid cells. IL-12 likely comes from myeloid cells and IFNg and IL-17 probably come from T cells and ILCs, but not epithelial cells as is written.
Correct. The sentence has been reformulated.
Reviewer 2 Report
The manuscript (text and figures) has been significantly improved and questions adequately addressed.
Author Response
Thank you